# Feature-Based Occupancy Map-Merging for Collaborative SLAM

**DOI:** 10.3390/s23063114

**Published:** 2023-03-14

**Authors:** Sooraj Sunil, Saeed Mozaffari, Rajmeet Singh, Behnam Shahrrava, Shahpour Alirezaee

**Affiliations:** Department Electrical and Computer Engineering, University of Windsor, Windsor, ON N9B 3P4, Canada

**Keywords:** collaborative SLAM, occupancy grid maps, feature-base map merging, heterogeneous on-board sensors

## Abstract

One of the most frequently used approaches to represent collaborative mapping are probabilistic occupancy grid maps. These maps can be exchanged and integrated among robots to reduce the overall exploration time, which is the main advantage of the collaborative systems. Such map fusion requires solving the unknown initial correspondence problem. This article presents an effective feature-based map fusion approach that includes processing the spatial occupancy probabilities and detecting features based on locally adaptive nonlinear diffusion filtering. We also present a procedure to verify and accept the correct transformation to avoid ambiguous map merging. Further, a global grid fusion strategy based on the Bayesian inference, which is independent of the order of merging, is also provided. It is shown that the presented method is suitable for identifying geometrically consistent features across various mapping conditions, such as low overlapping and different grid resolutions. We also present the results based on hierarchical map fusion to merge six individual maps at once in order to constrict a consistent global map for SLAM.

## 1. Introduction

An accurate spatial model of the environment and the relative position of the robot are essential for robotic perception, especially for mobile autonomous systems. In robotics, these two essential components are estimated jointly when given the on-board sensory observations. The process of estimating the robot pose and the environment map are popularly known as simultaneous localization and mapping (SLAM) [1]. The solution to the SLAM problem is considered to be one of the fundamental requirements to achieve true robot autonomy. The SLAM could be extremely useful during scenarios when the global positioning system (GPS) is inaccessible, e.g., in environments such as indoor [2], drone [3], underwater [4], forest canopy [5], medical [6], etc. However, the interdependence nature of localization and mapping makes SLAM a complex and challenging problem in practical applications. Despite its complexity, SLAM has been a theoretically well-studied field of research that still draws significant attention due to its vast practical importance.

While single-robot SLAM is challenging enough, extending it to multiple robots introduces a new set of problems [7,8], among which, a major challenge is to incorporate all the available information obtained from individual robots into a common global reference frame if the initial poses were known [9,10]. When the initial relative poses of robots are known, the multi-robot map-merging problem can be utilized as an extension of the single-robot scenario [11]. However, in many situations, the assumption of knowing the initial relative position of the robot can hardly be satisfied. Many studies have focused on situations in which the initial poses are unknown [12,13]. At present, the research on SLAM has shifted from the indoor environment to outdoor environments or to large-scale complex dynamic scenarios. Thus, it is becoming more difficult to complete the SLAM task using only a single robot. To overcome this issue, more attention is now being paid to multi-robot SLAM due to its high-efficiency [14,15,16], which involves more than one robot working collaboratively in a certain scenario. Collaborative SLAM can significantly improve the map efficiency in terms of the map quality and construction time. The quality of the maps generated by individual robots varies due to the differences in their on-board sensors, making global maps construction more challenging.

For multi-robot SLAM, three tasks are required: the determination of the relative poses; data exchange; and map merging, as shown in Figure 1. Robot relative poses are crucial for exploring the environment map using multiple robots [17]. Technologies such as Bluetooth [18], RIFD, and WIFI [19] can be used to communicate between the robots for data exchange. Two key steps involved in map fusion are map alignment and data association [20]. The map alignment process determines the appropriate spatial coordinate transforms between local maps. Many map alignment algorithms have been developed based on some factors, such as similar map formats, the map scale, grid resolution, and map overlapping ratio [21,22,23,24,25,26]. The purpose of the data association is to match and merge the map features between local maps to generate the global map established by multiple robots.

In general, the feature correspondence is required to calculate the overlapping area between the local maps generated by the individual robots. A major drawback in such methods is that it usually fails to find a sufficient number of robust feature correspondences. In this paper, we propose a new map merging method based on KAZE feature descriptor [27]. Our novelties include: Processing the spatial occupancy probabilities and detecting features based on locally adaptive nonlinear diffusion filtering.Developing a procedure to verify and accept the correct transformation to avoid ambiguous map merging.Proposing a global grid fusion strategy based on the Bayesian inference, which is independent of the order of merging.

The consistency and effectiveness of the proposed method are tested experimentally using real-world data. The results test the applicability of the method for merging maps with low overlapping and at different grid resolutions. The rest of the article is organized as follows: Section 2 reviews the existing literature on collaborative mapping in detail; Section 3 formulates the map fusion problem; Section 4 describes the proposed solution; Section 5 presents the effectiveness of the presented approach based on the real-world experimental results; finally, the conclusions are drawn in Section 6.

## 2. Related Literature

The initial experimental results [27] based on manual feature extraction depicted the potentiality of feature-based techniques to perform successful map merging. Since then, numerous methods based on feature matching have been proposed to solve the collaborative mapping problem. In this section, the state-of-the-art map merging approaches that utilize geometric features, specifically for the case of occupancy grid maps, will be exclusively reviewed.

The occupancy grid map of an indoor environment typically consists of distinguishable features, such as corners, edges, columns, or doorways. This is also true for specific outdoor maps, which consist of walls, buildings, vehicles, etc. As the environment is assumed to be static, the features are well defined in an occupancy map. Hence, the grid maps can interpret gray scale raster images and image registration techniques can be employed to solve the map fusion problem. The overall steps involved in map image registration can be divided into the following stages [28]:Feature (keypoint) Detection: during this stage, the map image is searched for locally distinctive locations that are likely to match with other images.Feature Description: the region around each detected feature is converted into a compact and stable descriptor that can be used to match against other descriptors.Feature Matching: finally, at this stage, we efficiently search for likely matching candidates between two set of descriptors to establish the pair wise correspondence.

Figure 2 depicts a simplified schematic of the occupancy grid map fusion based on feature detection and matching. This technique can be straightforwardly extended to *n* number of robots where the computation increase based on their combinations is *C*(*n,k* = 2). A probabilistic approach to the map merging problem was formulated using the multi-hypothesis Random sample consensus (RANSAC) [29] and the iterative closest point (ICP) algorithms. To select the best feature detector, the performance of the scale invariant feature transform (SIFT) [30], speeded up robust features (SURF) [31], and Kanade-Lucas-Tomasi (KLT) [32] were studied in this paper. The results mainly highlighted the distinctiveness of the different feature detectors and descriptors in terms of image filtering, minimum error, repeatability, and computation time.

Specifically, it was shown that the different feature detectors required different filter sizes (Gaussian filter and median filter) to achieve their best performance. The SIFT feature detector was also used in [33] to detect the overlapping regions across maps. In this approach, the scale invariant property of the SIFT detector was exploited to merge maps with different resolutions. The problem of map fusion is viewed as a point set registration problem in [34,35]. These approaches used a variant of the ICP algorithm to find the required transformation to align the overlapping maps. The ICP algorithm is local convergent, hence it requires good initialization. Therefore, in [34,35], the image feature matching using SIFT was used to provide the initial transformation. Lin et al. [34] used the Harris corner detector to extract the edge point set. In [34], a multi-scale context-based descriptor based on the eigen values and eigenvectors was designed to describe the detected edge features. All three approaches [27,34,35] used the RANSAC algorithm to find geometrically consistent feature correspondences. Then, the scaling trimmed iterative closest point (STrICP) [35] was used to perform the point set registration to refine the transformation. Additionally, a robust motion averaging was used to recover the global transformation from a set of relative transformations obtained by matching numerous grid map pairs. In [33], maps were merged using the ORB (oriented features from accelerated segment test (FAST) and rotated binary a robust independent elementary features (BRIEF)) feature detector. The detected ORB keypoints were matched using the brute-force algorithm to establish putative feature matches. Afterward, the RANSAC algorithm was fitted to an affine transform model to estimate the required transformation to align the maps. A computationally fast corner detector for grid maps and a cylindrical descriptor [24] were used in [36] to enable collaborative SLAM. This method utilized the RANSAC algorithm to estimate and refine the transformation parameters. Further, it also presented a decision-making algorithm that reduces the overall exploration time. Although the SIFT-based methods are very popular, the Gaussian blurring involved in scale space construction may affect the feature localization at the pixel level [37]. As mentioned earlier, a major drawback in the feature-based methods is that it usually fails to find a sufficient number of geometrically consistent correspondences. Hence, many approaches [34,35] have used an additional refinement step to improve the final transformation. 

In addition to the above general feature detectors, some methods have been proposed specifically to find keypoints in the LIDAR data. Fast Laser Interest Region Transform FLIRT) [38,39] works by detecting the regions of the image that have high gradient magnitudes and orientations. FLIRT then computes a local descriptor for each of these keypoints based on the gradient orientations within the region. One of the main advantages of FLIRT is its speed. It can detect and describe keypoints in an image in real-time, making it useful for real-time applications, such as SLAM. However, detecting keypoints with high gradient magnitudes makes the FLIRT descriptor sensitive to sensor calibration. To tackle this issue, the Fast Adaptive LRF Keypoint Extractor (FALKO) feature detection algorithm was proposed [40]. It uses a local reference frame (LRF) to define the orientation of the keypoints and has been shown to outperform other popular feature detectors, such as SIFT, in terms of speed. Although being fast and robust to noise, FALKO may miss some corners and edges, generating fewer keypoints. Binary Robust Independent Elementary Features (BRIEF) is another feature descriptor [41]. It represents each keypoint as a binary string instead of a vector of real-valued numbers. This makes it computationally efficient and allows it to be used in real-time applications. Its limited discriminative power is one of the BRIEF descriptor’s disadvantages. It may not be as discriminative as those generated by other algorithms, which can make it more difficult to match features across different images.

Overall, there is no one-size-fits-all solution, and the choice of the keypoints detector depends on the specific application and requirements of the system. The factors to consider include the accuracy, computational efficiency, and robustness to environmental conditions such as noise or lighting. The presented map registration combination in this article results in a suitable number of feature matches, avoiding the need for an iterative transformation refinement stage [42]. For the validation of the correct map transformation, we present a procedure to verify and accept the correct transformation to avoid ambiguous map merging.

## 3. Problem Formulation 

In order to formulate the collaborative SLAM problem, consider the state sequence $x_{1:t}=\left\{x_{1},x_{2,\;\ldots \ldots \ldots }x_{t}\right\}$ of the robot that comprises the pose (position and orientation) information. The sequence $z_{1:t}=\left\{z_{1},z_{2,\;\ldots \ldots \ldots }z_{t}\right\}$ denotes the sensory observations made by the robot. The input control (or odometry) sequence $u_{1:t}=\left\{u_{1},u_{2,\;\ldots \ldots \ldots }u_{t}\right\}$ accounts for the change in *x* within the environment. Now, the goal of the SLAM problem is to recover a model of the surrounding environment $m_{t}$ and the sequence $x_{1:t}$ given $u_{1:t}$ and $z_{1:t}$. Based on this terminology, the posterior of collaborative SLAM involving two robots *a* and *b* based on the direct observations can be denoted as follows [1]:(1)$$p\left(x_{1:t}^{a},x_{1:t}^{b},m_{t}|z_{1:t}^{a},z_{1:t}^{b},u_{1:t,}^{a}u_{1:t,}^{b}x_{0}^{a},x_{0}^{b}\right)$$
where $x_{0}$ indicates the initial pose information. Figure 3 graphically depicts the direct collaborative SLAM problem involving two robots. In order to formulate the indirect solution (i.e., map fusion), consider two local maps *^a^m* and *^b^m* generated by robots *a* and *b*. For simplicity, the time index *t* is disregarded. Assuming that the individual maps are modeled as occupancy grids, the grids can be viewed as *a*(*x*, *y*) and *b*(*x*, *y*) map images. Therefore, the pixel co-ordinates of the maps can be denoted as:(2)ma=[axay] and mb=[bxby]

Further, if there is overlapping between the maps, the motive is to find the geometric transformation that transforms [*^b^x*, *^b^y*] → [*^a^x*, *^a^y*] in such a way that the overlapping areas fall squarely on top of each other.

*T* transforming mb to ma using transformation. *T* can be written as follows:(3)mba=Tba(mb)

Based on the resolutions ra, rb of the individual grid maps (ra, rb), the transformation *T* can take two models. If ra≠rb, then *T* takes the similarity transform model for which the transformation parameters, such as translation *l*, rotation $R_{\theta }$, and scale change λ, need to be estimated. Given the estimated parameters, the map *m* can be transformed into $\bar{m}$ as [1]:(4)$$\bar{m}=\lambda Rm+l$$
(5)$$\lambda =\left[\frac{\lambda_{x}}{\lambda_{y}}\right],R_{\theta }=\left[\begin{array}{cc} \cos\theta & -\sin\theta \\ \sin\theta & \cos\theta \end{array}\right],\mathrm{and}\;l=\left[\frac{l_{x}}{l_{y}}\right]$$

If the grid maps are constructed using the same scale (i.e.,ra=rb), then the scaling factor *λ* = 1. In this case, *T* reduces to a special instance of similarity transformation which is the rigid transform. Therefore, Equation (4) can be simplified as: (6)$$\bar{m}=Rm+l$$

It must be noted that during occupancy grid mapping, the resolutions of the maps are known. Hence, even for the case of heterogeneous resolutions, only the rotation and translation will be estimated. For convenience, the transformation is represented using the homogeneous coordinates as:(7)$$T\left(l_{x},l_{x},\theta \right)=\left[\begin{array}{ccc} \cos\theta & -\sin\theta & l_{x}\\ \sin\theta & \cos\theta & l_{y}\\ 0 & 0 & 1 \end{array}\right]$$
where *T* is the map transformation matrix and it is generally estimated by pairing two grid maps. Figure 4 depicts the indirect collaborative map merging problem involving i=1,……‥,n robots. It is important to note that each robot maintains its own individual SLAM map.

## 4. Proposed Method 

In this section, we present our map fusion method in detail. The overall steps involved in estimating the transformation *T* to align the maps is summarized as follows: the probability layer of a grid map is systematically processed to separate the high probable grid information. This allows us to perform accurate feature detection and matching between the maps. Then, the KAZE feature [37], which utilizes locally adaptive nonlinear diffusion filtering, is used to construct the scale space. The choice of interest feature is different from the previous feature-based map merging methods [27,43]. Afterwards, the nearest-neighbor matching technique is implemented to find the putative correspondences between the detected features. The RANSAC algorithm is utilized to obtain the map transformation matrix by eliminating outliers in the putative correspondence. Finally, the verification procedure based on the map acceptance index and the cardinality of the largest inlier set to accept a valid transformation is deployed. Figure 5 illustrates an example cycle of successful map registration between two laser-based grid maps with reasonable overlaps using the proposed method.

### 4.1. Processing Occupancy Maps

Most of the existing map merging techniques represent the occupancy maps as map images by sampling the occupied cells to an almost uniformly distributed point set [36]. However, grid maps usually consist of artifacts that can be viewed as high-frequency noises in images. Hence, some approaches have applied image filters (blurring), which may affect the pixel level accuracy of the keypoint localization. To tackle this problem, instead of using the occupied cells, we use the obstacle free cells. This has two main advantages: firstly, the number of obstacle-free cells in a map is greater compared to the occupied cells; secondly, the obstacle-free layer will usually have higher grid probabilities. By exploiting these two general assumptions, we can apply a global threshold to the obstacle-free layer of an occupancy map to eliminate cells with lower probabilities.

The obstacle-free cells in a probabilistic occupancy map can be easily extracted by exploiting the cell-independence assumption. To achieve this, consider the probability layer of an occupancy map *m* as a matrix *M* with *r* rows and *c* columns. Each grid cell in *M* denotes a binary random variable that indicates the probability of occupancy $p\left(m_{i}\right)$ of the cell. The index *i* indicates the pixel coordinate position $i={\left[x,y\right]}^{T}$, where x=1,……‥,r and y=1,……‥,c. The occupancy information, such as whether a cell is occupied, obstacle-free, or unknown, can be inferred based on the probability values, as:(8)$$p\left(m_{i}\right)=\left\{\begin{array}{c} 1\\ 0\\ 0.5 \end{array}\begin{array}{c} \mathrm{then}\\ \mathrm{then}\\ \mathrm{then} \end{array}\begin{array}{c} p\left(m_{i}=occ\right)\\ p\left(m_{i}=free\right)\\ p\left(m_{i}=unknown\right) \end{array}\right\}$$

We are interested in $p\left(\neg m_{i}\right)$, which is the probability of free cells in the map. Assuming $p\left(m_{i}\right)=1-p\left(\neg m_{i}\right)$, we can easily calculate the probability of the *i^th^* cell being obstacle-free as:(9)$$p\left(m_{i}=free\right)=p\left(\neg m_{i}\right)=1-p\left(m_{i}\right)$$

Figure 6 illustrates an exemplar occupancy grid map alongside the extracted occupied and obstacle-free probability layer. It can be clearly observed that the obstacle-free layer is more informative than the occupied layer as it has high probable spatial information. It also preserves the structural boundaries of the environment. Now, the structural details presented in the free layer can be refined by removing the cells (setting to 0) with low probability values $p\left(\neg m_{i}\right)$. For example, we can set all values of $p\left(\neg m_{i}\right)$ to 0, expect for $p\left(\neg m_{i}\right)$ *> α*, where 0.01 *≤ α ≤* 0.99 is a user-controlled threshold factor. Let the map image after thresholding be denoted as *I*. Then, *I* will only consist of obstacle-free cells whose probability is greater than *α*. That is, the values of the occupied and unknown cells in the map will be 0. This simple but effective manipulation of the grid maps is highly utilitarian in the process of feature detection and matching. 

### 4.2. Feature Detection 

A keypoint, in general, is a locally distinctive point (also region or line) in an image based on the nature of the intensity values around it. Repeatable keypoints can be identified using keypoint detection methods. Classical 2D feature detectors construct the scale space of an image by filtering the original image with an appropriate function over the increasing time or scale. For instance, the Gaussian scale space [30] is constructed by convolving the image with a Gaussian kernel of increasing standard deviation. With multi-scale image representation, image features can be detected at different scale levels or resolutions. However, Gaussian blurring filters both details and noise to the same degree [43]. This adversely impacts the localization accuracy of the detected keypoints [37].

#### 4.2.1. Nonlinear Diffusion Filtering 

Whenever discrete images are viewed as continuous objects, the powerful calculus of variations can be applied. This enables us to perform filtering that depends on the local content of the original image. The nonlinear diffusion enables locally adaptive filtering, which preserves the crucial edge details in the occupancy images. The filtering can be described by the following partial differential equation:(10)$$\frac{\partial I}{\partial t}=div\left(c\left(x,y,t\right)\cdot \nabla I\right)$$
where div and ∇ are the divergence and gradient operators, respectively, and *I* is the input image. The conductivity function *c* enables the diffusion to be locally adaptive to the image structure. The Perona-Malik diffusion in [36] proposed to set the function *c* to be dependent on the gradient magnitude, as:(11)$$c\left(x,y,t\right)=g\left(\left|\nabla I_{\sigma }\left(x,y,t\right)\right|\right)$$
where $\nabla I_{\sigma }$ is the gradient of a Gaussian smoothed version of the original image *I* with the standard deviation *σ*. Here, we use the following diffusion coefficient *g* from [36]:(12)$$g=\frac{1}{1+\frac{|\nabla I\sigma {|}^{2}}{c^{2}}}$$
where the parameter *c* is the contrast factor that controls the sensitivity to edges. The value for *c* is usually estimated or made as a function of the noise in the image. Further, with the constant diffusion coefficient, the diffusion equations reduce to the heat equation, which is equivalent to Gaussian blurring. 

As there is no analytical solution to the partial differential equations (PDEs), the diffusion in Equation (10) should be approximated. Here, we use exact the discretization of the KAZE features presented in [37], which adopts a semi−implicit scheme. The discretization can be expressed as:(13)$$\frac{I^{i+1}-I^{i}}{\tau }=\sum_{l=1}^{m}A_{l}\left(I^{i}\right)I^{i+1}$$
where $I^{i}$ and $I^{i+1}$ are the filtered images at the current and next level, respectively; *τ* is the time difference and $A_{l}$ is a matrix that encodes the image conductivity for each dimension. The solution for $I^{i+1}$ can be obtained as follows:(14)$$I^{i+1}={\left(I-\sum_{l=1}^{m}A_{l}\left(I^{i}\right)I^{i+1}\right)}^{-1}I^{i}$$

#### 4.2.2. KAZE Features

To detect the keypoints of interest, the nonlinear scale space is constructed using the scheme and variable conductance diffusion described in the previous subsection. A similar implementation to SIFT without sub-sampling is adopted to build the scale space arranged in logarithmic steps in the series of *O* octaves and *S* sub-levels. The octave and the sublevel indexes are mapped to their corresponding scale *σ* as:(15)$$\sigma_{i}\left(o,s\right)=2^{o+s/2},o\in \left[0\ldots O-1\right],\;s\in \left[0\ldots S-1\right],i\in \left[0\ldots K\right]$$
where *o* and *s* are the discrete octave and sublevel indexes. The values *O* and *S* show the maximum ranges of *o* and *s*. Parameter *K* is the total number of filtered images. As nonlinear diffusion filtering operates in time units, the scale level is mapped $\sigma_{i}\to t_{i}$ to convert pixel to time units as:(16)$$t_{i}=\frac{1}{2}\sigma_{i}^{2},i=\left\{0\ldots .K\right\}$$

Once the nonlinear scale space is constructed, the interest points are detected based on the response of the scale-normalized determinant of the Hessian Matrix at multiple scale levels. The determinant at different *σ* scale levels can be computed as:(17)$$I_{Hessian}=\sigma^{2}\left(I_{xx}I_{yy}-I_{xy}^{2}\right)$$
where $I_{xx}$, $I_{yy}$ are the second order horizontal and vertical derivatives, respectively, and $I_{xy}$ is the second order cross derivative. The derivatives are approximated by 3 *×* 3 Scharr filters and the maximum is computed at all scale levels, with the exception of the *I* = 0 (first) and *I* = *N* (last) scales. The main orientation in a local neighbourhood is obtained through the derivatives around each keypoint within a radius of $6\sigma_{i}$, where $\sigma_{i}$ is the sampling step. Each derivative in the circular area is weighted with a Gaussian centered at the keypoint. For each point, the dominant orientation is calculated by summing the derivatives within a sliding circle window over an angle of $\frac{\pi }{3}$.

### 4.3. Feature Description 

The descriptor is a finite vector that summarizes the properties, such as location, orientation, and response strength, of the detected keypoints. In this article, the SIFT descriptor [30] is used to describe the detected features. This is mainly due to the slight distinctiveness shown by SIFT [39] while describing the features. Moreover, the KAZE descriptor [37] is an extension of the SURF description method [31] to the nonlinear scale space. Therefore, using the KAZE descriptor may also result in good matching results.

### 4.4. Feature Matching 

Once the set of features are extracted, the feature matching stage sets up a putative pairwise correspondence between the detected keypoints. The feature matching problem can be formulated as follows: given the feature descriptor $f_{a}^{1:n}=\left\{f_{a}^{1},\ldots \ldots .,f_{a}^{n}\right\}$, we seek the nearest neighbor to $f_{a}$ from set $f_{b}^{1:m}=\left\{f_{b}^{1},\ldots \ldots .,f_{b}^{m}\right\}$. The matching process usually requires an exhaustive search, and it is the most time-consuming stage. In this article, the pairwise Euclidean distance *d* between the two feature vectors is used to find the nearest neighbor. The sum of the squared differences (SSD) is used as the matching metric and the distance between two feature vectors $f_{i}$ and $f_{j}$ is calculated as:(18)$$d\left(f_{i},f_{j}\right)=\sum_{u}\sum_{v}{\left(f_{i}\left(u,v\right)-f_{j}\left(u,v\right)\right)}^{2}$$
where *u* and *v* are variable components along the x-direction and y-direction, respectively. It should be noted that ambiguous matches are rejected based on an effective ratio test [30]. The test accepts each match by comparing the closest neighbor with the second-closet neighbor. Therefore, only the unique matches below a certain matching distance are accepted as feature correspondence.

### 4.5. Keypoint Detectability 

Feature detectability can refer to the ability of a machine vision system to detect specific features or patterns in an image. This can be important in applications such as map merging for SLAM. The total number of keypoints detected by different methods are shown in Figure 7. The scanning rate was set to 2.5 Hz and the map resolution was increased from 8 cell/m to 20 cell/m. It must be noted that the occupancy matrix is generated without any prepossessing. It can be seen that the ORB detector finds the maximum number of features, whereas the SIFT detector extracts the minimum number of points. The KAZE detector is the runner-up in terms of keypoint detectability.

### 4.6. Outlier Elimination 

The matching described in the previous subsection can produce good matches (*inliers*). However, it may still be contaminated by inconsistent matches (outliers). Hence, for the next stage, a robust variant of the RANSAC algorithm [29] known as the M estimator Sample Consensus (MSAC) [40] algorithm is used to eliminate the outliers and extract the geometrically consistent *inliers*. RANSAC is a standard procedure to estimate the parameters of a certain mathematical model contaminated by outliers. The parameter model of interest here is the rigid transform in Equation (7). The iterative RANSAC steps to estimate the transformation *T* for the largest set of *inliers* from the set of matched pairs M{fai,fbj} is summarized in Algorithm 1.
**Algorithm 1**[T, inliers]=RANSAC(M{fai,fbj}, *N_trails_*) *inliers ⇐ *0*T ⇐ *0*n ⇐ *0**while** *n < N_trails_* **do***S ⇐ *Randomly select subset of samples with minimum number of correspondences   *T_h_ ⇐ * Hypothesize transformation for the minimal set*Inliers_h_ ⇐ *Test for number of consistent matches with *T_h_*          **if**
*inliers_h_ > inliers*
**then**                      *inliers ⇐ inliers*
**then***T ⇐ T_h_***end if****end while**

The RANSAC algorithm accepts feature pairs as *inliers* if it lies within a threshold *τ*. The experimental results show that for higher *τ* values, the algorithm may produce a poor estimate. This is mainly due to the cost function *C*, which is used to score the *inliers* and outliers. To avoid this drawback, the statistically robust MSAC is used. The MSAC algorithm introduces a new error term for the cost function into the RANSAC algorithm. The cost function *C* is defined as [44]:(19)$$C=\sum_{i}\rho \left(e_{i}^{2}\right)$$
where the error term *ρ* for the RANSAC and the MSAC algorithms are as follows:(20)$$\rho_{RANSAC}\left(e_{i}^{2}\right)=\{\begin{array}{c} 0\\ cons\tan t \end{array}\begin{array}{c} e^{2}<\tau \\ e^{2}\geq \tau^{2} \end{array}$$
(21)$$\rho_{MSAC}\left(e_{i}^{2}\right)=\{\begin{array}{c} e^{2}\\ \tau^{2} \end{array}\begin{array}{c} e^{2}<\tau \\ e^{2}\geq \tau^{2} \end{array}$$

The main difference between Equations (20) and (21) is that the $\rho_{RANSAC}$ scores each outlier with a constant penalty, whereas $\rho_{MSAC}$ scores a penalty for the outlier as well as the *inliers*. Thus, the MSAC algorithm yields an added benefit with no additional computation burden. Figure 8a illustrates the location of the geometrically consistent KAZE features present in the common obstacle-free (described in Section 4.1) regions across two grid maps for one run of the MSAC algorithm. Figure 8b shows the cardinality of the largest set of *inliers* returned by the MSAC algorithm for different feature detectors. The effect of the thresholding cell probabilities based on *α* can be clearly observed, and the best result was produced by the KAZE feature detector and SIFT descriptor with *α* = 0.99 (i.e., by considering only the free grid cells whose probability is *≥*0.99). It should be noted that when *α* = 0, the original map itself is considered for feature matching. Thus, Figure 8b also highlights the resulting improvements by the processing stage described in Section 4.1. Clearly, all of the detectors and descriptors produced similar results for a low value of *α*. However, as *α* is increased, the structural details present in the map become more refined. This refinement considerably supports the locally adaptive filtering i.e., the KAZE features. Hence, the KAZE feature detector outperforms all the other methods, including the SIFT detector.

### 4.7. Grid Fusion

The final stage of the map fusion process is to verify the estimated transformation; then, fuse the pairwise local probabilistic cell information of the individual maps to build a consistent global map of the environment.

#### 4.7.1. Transformation Verification

We use the following two conditions to accept the estimated transformation:Although only two valid feature correspondences are sufficient to estimate the transformation, it is highly unlikely that the correspondences are true positives. Hence, only the transformation for minimum inlier cardinality (well-over two feature correspondences) is accepted.Further, we use the acceptance index based on pairwise cell agreement and disagreement between the map matrix M and the transformed map matrix $\bar{M}$ to check the quality of the transformation. The acceptance index $\omega \left(M,\bar{M}\right)$ is defined as:(22)$$\{\begin{array}{c} 0\\ \frac{agr\left(M,\bar{M}\right)}{agr\left(M,\bar{M}\right)+dis\left(M,\bar{M}\right)} \end{array}\begin{array}{c} agr\left(M,\bar{M}\right)=0\\ agr\left(M,\bar{M}\right)\neq 0 \end{array}$$
where $agr\left(M,\bar{M}\right)$ indicates the number of cells in M and $\bar{M}$ that agrees (either free or occupied). The disagreement $dis\left(M,\bar{M}\right)$ is the number of cells that are different (either M is free and $\bar{M}$ is occupied, or vice-versa).

#### 4.7.2. Certainty Grid Fusion

Once the transformation is verified, the next stage is to combine the grid probabilities of the two local maps. Many existing methods focus only on finding the transformation and aligning the maps, whereas the fusion problem is given less consideration. The local grid probabilities were simply added by exploiting the additive property of log-odds [7]. In [26,27], the grid fusion rules were defined as lookup tables.

In this article, we use the Bayesian inference to deal with the uncertainty and update the global grid probabilities. Using Bayes’ rule, the probability of the global grid p({mi}G), given the grid probability p({mi}a) of map *a* and the transformed (b→a) grid probability p({mi}ab) of map *b*, can be calculated as:(23)p({mi}G|{mi}a,{mi}ab)=ABAB+(1−A)(1−B)
where B=p({mi}G|,{mi}ab). The grid fusion in Equation (23) can be extended to *n* robot maps, as follows:(24)1p({mi}G|{mi}1,……{mi}n)−1=∏k=1n(1p({mi}G|{mi}n)−1)

As both Equations (23) and (24) are associative and commutative, the order of operations is not important.

#### 4.7.3. Transformation Reliability

The results of the map matching may not be exactly identical because of the randomized nature of the MSAC algorithm. Therefore, the RMSE [45] is used to compare the reliability of the different feature detectors based on the alignment error. The RMSE is generally used to measure the difference between the estimated and the true parameter of interest. Here, it is used to compare the different feature detectors based on their alignment error caused by rotation. The RMSE of MSAC with respect to the rotation *θ* can be calculated as follows:(25)$$\mathrm{RMSE}=\sqrt{\frac{1}{N}\sum_{n=1}^{N}{\left(\theta -\hat{\theta }\right)}^{2}}$$
where $\hat{\theta }$ indicates the estimated rotation parameter and *N* denotes the number of Monte Carlo runs. The reason for only using the rotation, but not the translation, is that the rotation component of the transformation is the hardest to recover [25].

In order to compute the RMSE, knowledge about the true rotation *θ* is required. Hence, the error is computed by applying random rotation to the occupancy maps and recovering the applied rotation. The estimated rotation $\hat{\theta }$ can be recovered as
(26)$$\hat{\theta }=a\tan2\left(\lambda \sin\hat{\theta },\lambda \cos\hat{\theta }\right)$$
where λ is the scaling factor. Figure 9 illustrates the RMSE calculated over 100 runs for matching maps with 100 % overlap. For each rotated source map, the $\hat{\theta }$ was calculated based on Equation (26). The average time for each feature detector and descriptor was calculated as:*Avg. time* = *t*_detection_ + *t*_description_ + *t*_matching_ + *t*_MSAC_(27)
where *t*_detection_, *t*_description_, *t*_matching_, *t*_MSAC_ are the time taken for detecting the features, describing the detected features, matching the descriptors, and fitting the *inliers* using the MSAC algorithm.

On the whole, the KAZE method demonstrated minimal rotation errors compared to the other methods. However, it is the slowest in terms of computation time. The RMSE shown by the SIFT method was high, and it is also less reliable at times as it failed to estimate the rotation parameter between runs. The SURF and ORB methods were fast (SURF was fastest by a small margin), out of which the SURF had a better error performance.

## 5. Collaborative Mapping

In this paper, the local maps were generated using multiple robots (Qcars), as shown in Figure 10. The Qcars were remotely controlled, and the environment was scanned using the laser scanner of the individual Qcars. The 2-D Lidar is used for mapping the environment. The Rplidar A2M8 model is used by Qcar. The measuring range is between 0.2 m to 12 m, with a 0.45 degree resolution. The ROS platform is used to build the local maps using SLAM and the RPlidar library.

The SLAM incrementally processes the scans to build a pose graph that links the scans. Each node in the graph is connected by an edge constraint that relates the poses between the nodes and represents the uncertainty in that measurement. Whenever two regions of the map are found to be similar, the system calculates a relative transformation that aligns the two regions to close the loop. The algorithm utilizes the loop closure information to update the map and optimize the estimated pose trajectory. The inputs are two local maps ma and mb and the output is the global map mG. All of the map fusion computations were performed using the MATLAB/ROS software on a central CPU core processor, running at 3.60 GHz with 32 Gb of RAM. Two robots collaborate to build a map of an indoor environment. The map fusion stages are summarized in Algorithm 2.
**Algorithm 2** [mG]=map Fusion(ma; mb)        Process the maps to obtain occupancy images $I_{a}^{free}$, $I_{b}^{free}$        [Ia;Ib] = **process Occupancy Maps** (ma;mb)        Detect KAZE keypoints $k_{a}$, $k_{b}$        [$k_{a}$;$k_{b}$] = **detect KAZE features** (Ia;Ib)        Describe the detected features using the SIFT descriptor.         [$f_{a}$;$f_{b}$] = **SIFT description** (Ia;Ib;$k_{a}$;$k_{b}$)        Find the nearest-neighbors.        $M\left\{f_{a}^{i},f_{b}^{j}\right\}$ = **feature Matching** ($f_{a}$;$f_{b}$)        Compute the transformation T using the MSAC algorithm.        [**T**] = **outlier Elimination** ($M\left\{f_{a}^{i},f_{b}^{j}\right\}$)        Verify the transformation, and update the global map mG based on grid.         fusion methodology        [mG] = **grid Fusion** (ma;mb;*T*)

Figure 11 depicts the two overlapping local grid maps ma and mb with the same resolutions and the resulting merged global map $G_{m}$.

To merge maps at different grid resolutions, we require the scaling factor, in addition to the rotation and translation. Fortunately, the resolution of occupancy grid maps is a user-defined variable. Hence, it can be utilized to compute the scale change between the moving map (map to be merged) and the fixed map. Based on the scale change, the moving map can be rescaled to the same resolution as the fixed map using nearest neighbor interpolation. It is important to note that interpolating map images may alter the pixel level information. To attenuate the effect of interpolation, only the processed obstacle-free layer of the map is rescaled (i.e., the original map is not scaled). We estimate the parameters for the rigid transform matrix from Equation (7). Then, we define the similarity transformation in Equation (4), using the known scaling factor and the estimated rotation and translation. Finally, we apply the defined similarity transformation matrix to the original moving map even though we fit the rigid model to estimate the parameters. In Figure 12, the pixel locations of the occupied cells are overlaid for three resulting global maps. For all cases, the fixed map ma has a resolution of $r_{a}$ = 25 cells/m, whereas three moving maps mb have different resolutions: $r_{b}$ = 10, 20, 25 cell/m. It can be clearly observed that, for three cases, we obtained a consistent transformation to transform mb →ma. 

As stated before, due to the randomness involved in the MSAC algorithm, the estimated rotation and translation may vary between runs. Similarly, parameters such as the acceptance index *ω*, execution time, and the number of feature correspondences also vary. Hence, for a meaningful interpretation of the results, we summarize the performance of the map fusion in Table 1 by computing the average and the deviations over 100 runs. The number of iterations and the confidence for the MSAC algorithm were set to 1000 and 99%, respectively.

Table 2 shows the comparative results of both the MSAC and RANSAC algorithms for the same and different grid resolution maps. As depicted from Table 2, the performance of the MSAC algorithm is better than the RANSAC algorithm as the RANSAC algorithm may produce a poor estimate for higher threshold values.

### 5.1. Hierarchical Map Fusion

Whenever a set of maps (more than two) is available, the process of map fusion should be hierarchically executed. As feature-based map fusion techniques are fast, the hierarchy can be decided based on an exhaustive search. The maps obtained from the robots *a* and *b* and similarly *c* and *d* can be merged to find a transformation (if it exists) between *a*, *b*, *c*, *d*. Thus, robots in an interaction mode can form exploration clusters in which they can coordinate their actions. In this work, the lobby map of a large indoor environment is reconstructed using six local maps provided by individual robots. The motive is to implement the presented feature-based method in a real application. The individual robots explored the environment at different time intervals. This assumption is common when deploying multiple robots as they begin their operation individually. Each map differs from the other maps in terms of grid resolution. Further, a non-robust SLAM (map *a*) is included in the set to test whether the alignment algorithm can be performed correctly during the map fusion.

In the central computer, all six maps were provided at once. Then, the local maps were fused in a hierarchical fashion to construct the global map of the environment. Figure 13 depicts the executed hierarchy of merging. The map construction and merging order is as follows: map *a,* map *b*, map *c*, map *d*, map *e*, map *f →* {map *a*, map *b*}*,* {map *c*, map *e*}, {map *d*, map *f*} *→* {map *a*, map *b*, map *c*, map *d*, map *e*, map *f*}.

Finally, Figure 14 displays the obtained global map overlaid on the building blueprint to qualitatively interpret the alignment results. It must be emphasized that the single-robot SLAM was unable to construct the whole map of the environment, primarily due to accumulated errors. Furthermore, it consumed a large amount of memory and time to build the map.

After the verification of the global map generation, the simulation for the robot SLAM is conducted in the Robot operating software (ROS) environment via certain predefined waypoints, 1 to 14. The simulation results are presented in Figure 15. It shows the global map with waypoints and the path travelled by the robot during navigation in the global map. The ROS navigation package is used for moving the robot through defined waypoints. As depicted, the robot followed all of the waypoints in the global map environment.

### 5.2. Motion Planning

The next step after the collaborative map construction is motion planning, which aims to find a collision-free path for the provided start and end point location in the constructed map. This can be conducted by using the rapidly exploring random tree (RRT) algorithm [46]. The RRT algorithm samples random states within the state space and attempts to connect a trajectory. These states and connections are validated or excluded based on the map constraints. Figure 16 illustrates the path planning results computed using the two global maps obtained by the proposed map fusion method.

## 6. Conclusions

In this article, the unknown initial correspondence problem that arises in collaborative mapping systems was solved using the feature-based map registration method. Insufficient or inconsistent correspondences are one of the major drawbacks in feature-based map merging methods. We addressed this issue by exploiting the following: (i) capturing the spatial probabilistic information of the grid maps; (ii) detecting the keypoints based on locally adaptive nonlinear diffusion filtering (KAZE features); (iii) and describing the detected keypoints using the SIFT description. Another advantage of the presented approach is that it does not require any iterative transformation refinement stage, thus reducing the overall merging time.

We provided real-world map merging results for registering noisy, low overlapping, as well as different resolution, grid maps. Moreover, we implemented the presented map fusion approach in a hierarchical fashion to obtain a map of a large-scale indoor environment. The results highlight the importance of collaborative systems, as well as the feature-based merging techniques, which could be easily decentralized for practical applications.

For future work, the proposed feature-based map merging algorithm will be extended and validated for different multi-robot with different LIDAR specifications.

## Figures and Tables

**Figure 1 sensors-23-03114-f001:**
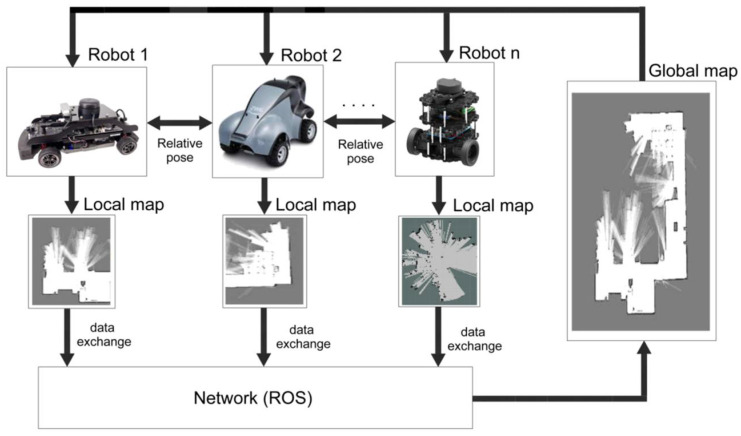
Collaborative simultaneous localization and mapping (SLAM).

**Figure 2 sensors-23-03114-f002:**
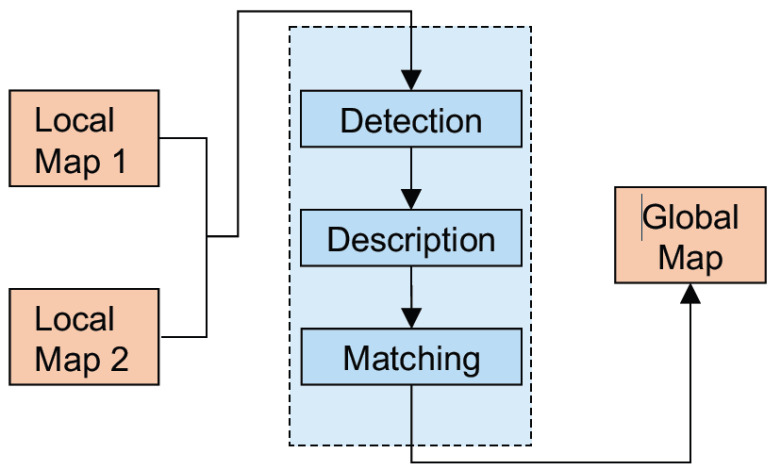
Map merging based on image feature matching.

**Figure 3 sensors-23-03114-f003:**
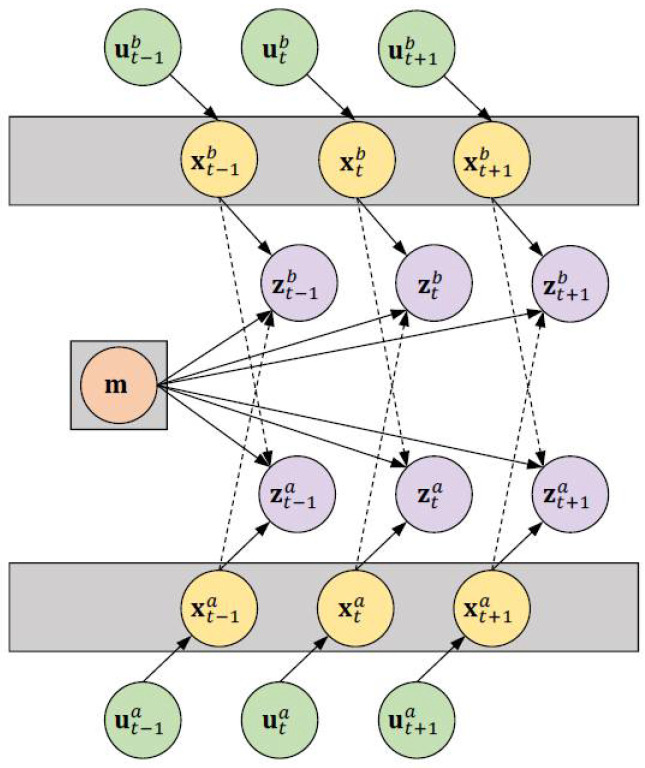
Graphical model of the collaborative SLAM problem for two robots *a* and *b*.

**Figure 4 sensors-23-03114-f004:**
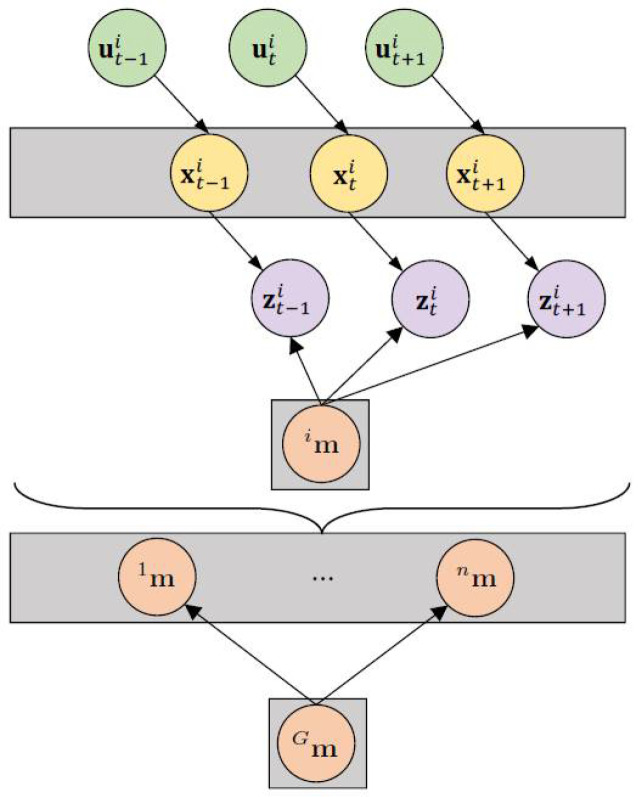
Graphical model of the map fusion problem (indirect solution) for *n* robots.

**Figure 5 sensors-23-03114-f005:**
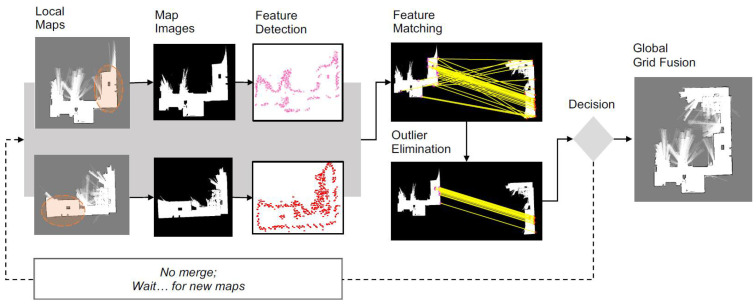
Cycle of the presented map fusion approach. Overlap between the maps is shown with a oval.

**Figure 6 sensors-23-03114-f006:**
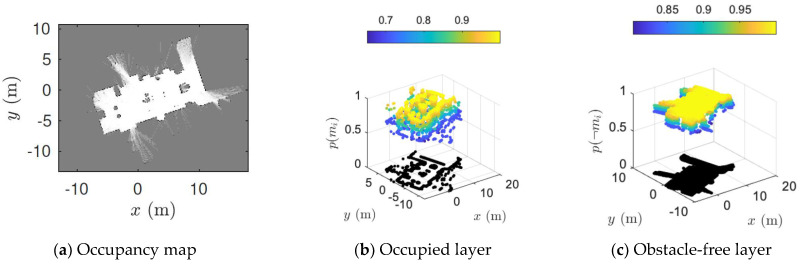
An example of probabilistic grid map alongside the position of occupied and obstacle-free cells with their respective spatial probabilities.

**Figure 7 sensors-23-03114-f007:**
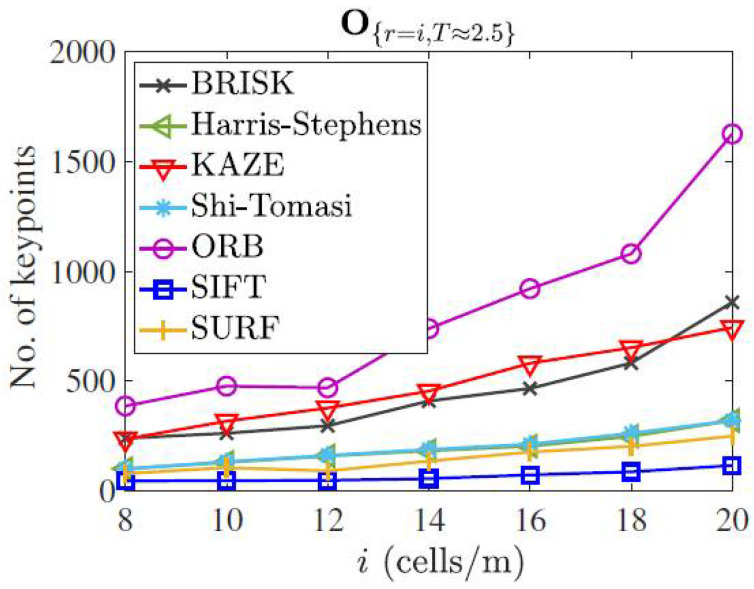
Extracted number of keypoints by different detectors in map *O*
_{*r,T*}_ for varying resolution (*r*) and scan rate (*T*).

**Figure 8 sensors-23-03114-f008:**
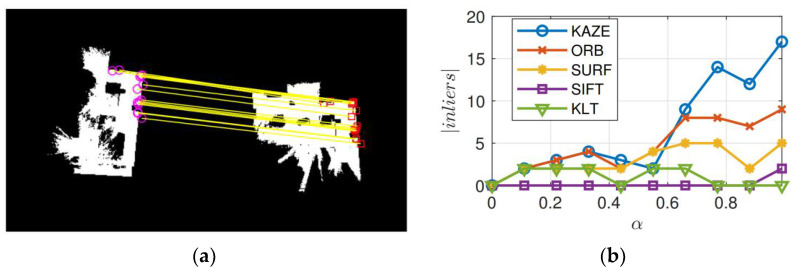
Geometrically consistent MSAC *inliers* across two occupancy maps. (**a**) Location of KAZE *inliers* in the processed obstacle-free layer with threshold *α* = 0.99, (**b**) Number of geometrically consistent features computed for KAZE, ORB, SURF, SIFT, and KLT methods.

**Figure 9 sensors-23-03114-f009:**
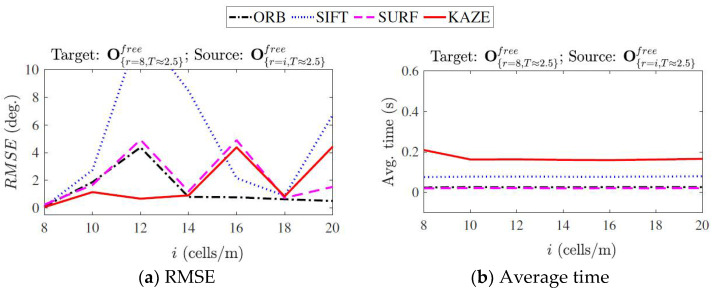
RMSE and average feature matching time for change in resolution computed over 100 runs.

**Figure 10 sensors-23-03114-f010:**
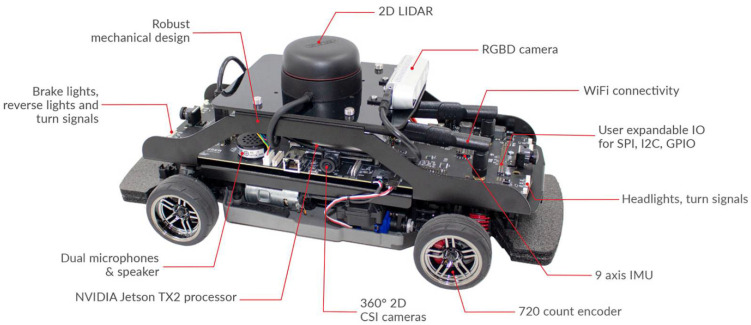
Qcar by Quanser.

**Figure 11 sensors-23-03114-f011:**
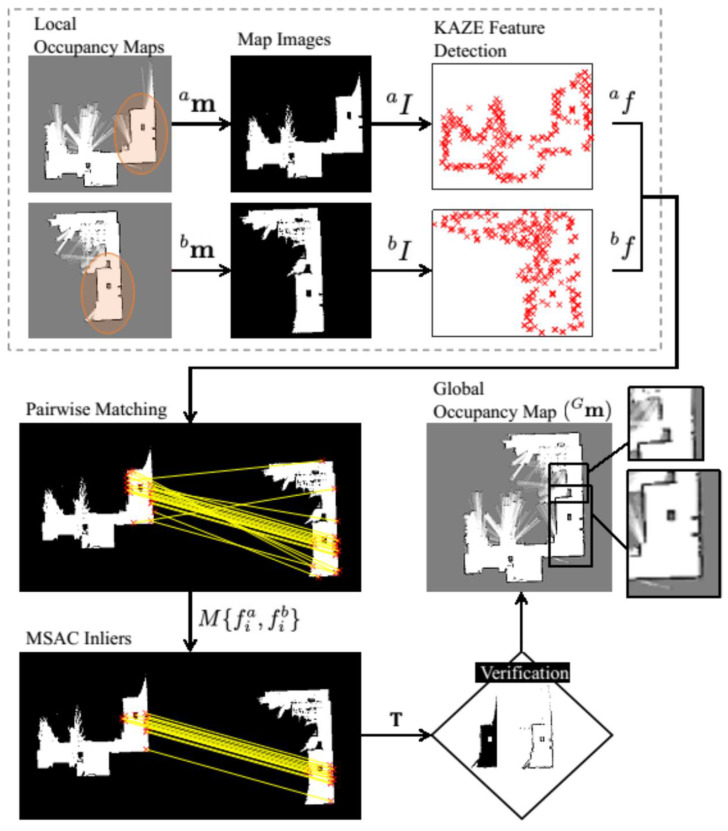
Feature matching technique using two maps with same grid resolutions ($r_{a}$ = $r_{b}$ = 20 cells/m).

**Figure 12 sensors-23-03114-f012:**
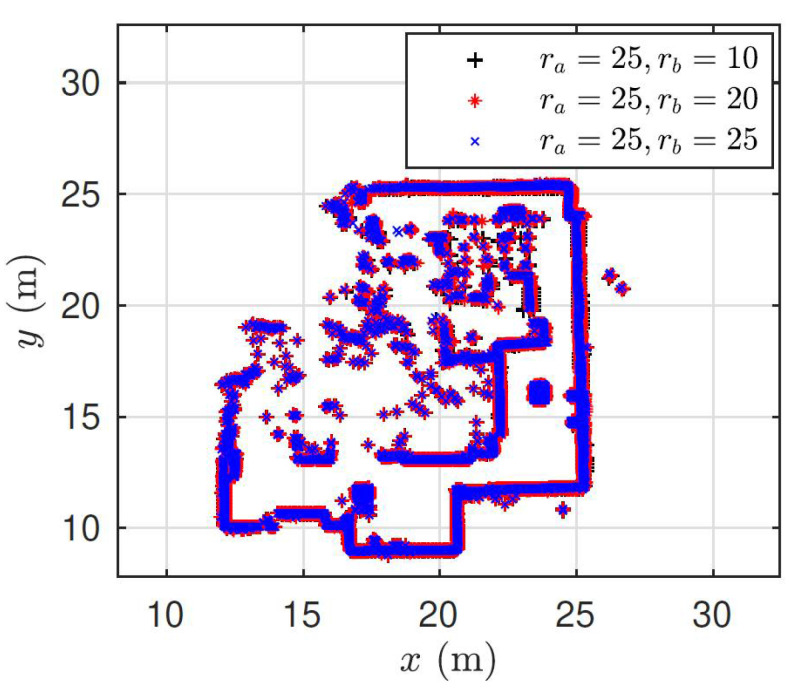
Occupied pixel locations of three global maps as a result of map fusion. Each map is obtained as a merging result between a fixed map ma of resolution $r_{a}$ = 25 cell/m and a moving map mb of resolutions $r_{b}$ = 10, 20, 25 cells/m.

**Figure 13 sensors-23-03114-f013:**
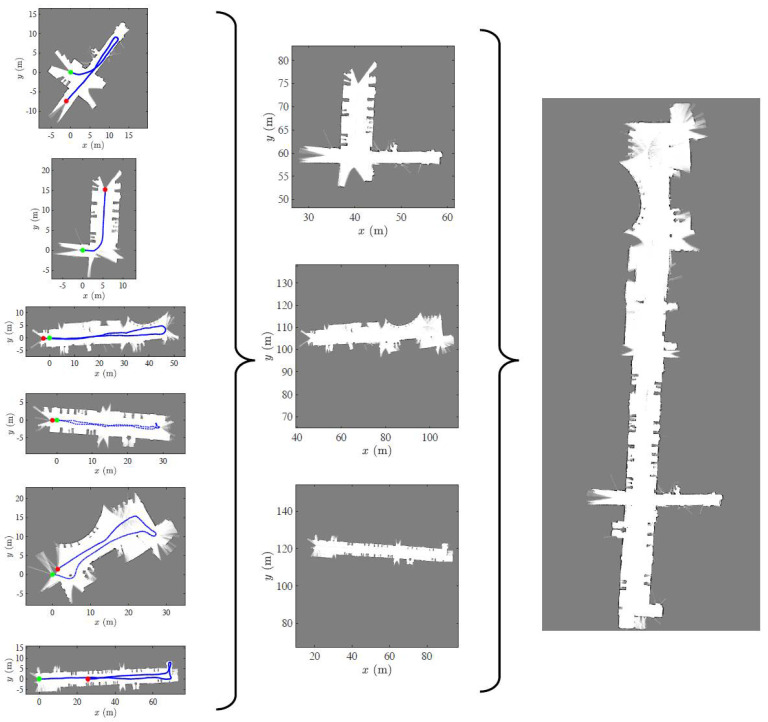
Map fusion hierarchy. The six heterogeneous local maps obtained from individual robots are hierarchically merged into a consistent global map.

**Figure 14 sensors-23-03114-f014:**
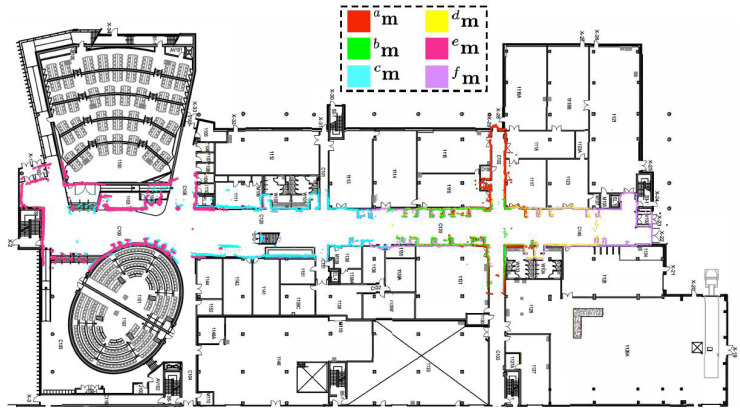
Map generated by individual robots overlaid on building blueprint.

**Figure 15 sensors-23-03114-f015:**
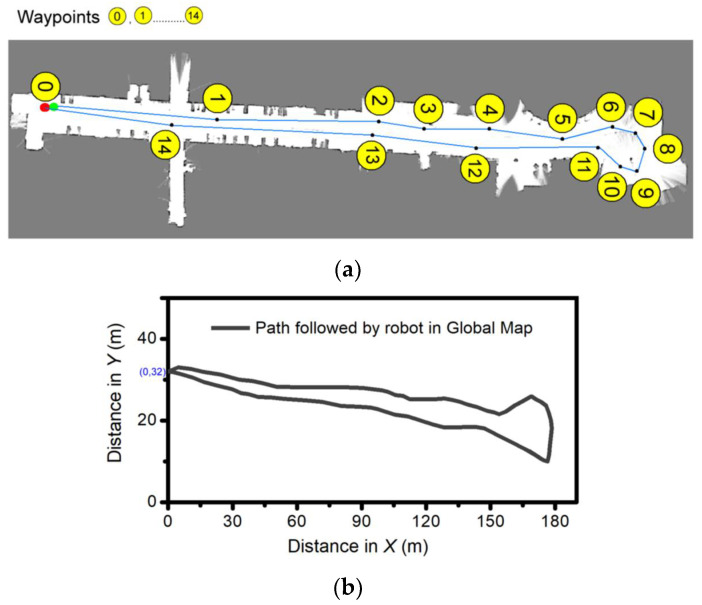
Simulation results: (**a**) Global map with waypoints, and (**b**) Path travelled by the robot during navigation in global map in ROS environment.

**Figure 16 sensors-23-03114-f016:**
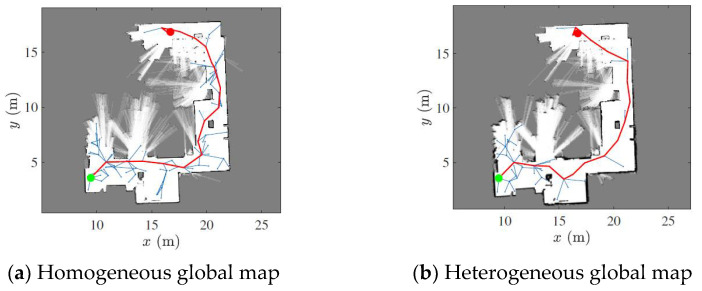
Path planning using the RRT algorithm and different global maps. Starting and end points are shown with red and green points. The robot path is marked with a red line.

**Table 1 sensors-23-03114-t001:** Mean and deviation of performance parameters while merging maps of same and different grid resolutions.

ma	mb	Gm(ma,Tab(mb))				
$r_{a}$ (Cells/m)	$r_{b}$ (Cells/m)	ω(ma,m¯b)	Wall-Clock Time (sec)	|*Inliers*|	Rotation (deg.)	Ranslation (m)
25	25	$0.98\pm 2.3\times 10^{-4}$	$0.49\pm 7.7\times 10^{-3}$	33	±0.0472	±0.9274
20	20	$0.97\pm 5.1\times 10^{-4}$	$0.32\pm 4.9\times 10^{-3}$	35	±0.0935	±1.4873
10	10	$0.95\pm 2.0\times 10^{-3}$	$0.20\pm 2.3\times 10^{-3}$	21	±0.1011	±1.7138
25	20	$0.97\pm 7.8\times 10^{-4}$	0.48±0.0113	28	±0.1046	±1.5863
25	10	$0.93\pm 9.8\times 10^{-4}$	$0.47\pm 7.2\times 10^{-3}$	25	±0.2161	±1.6717
20	10	$0.94\pm 3.2\times 10^{-3}$	$0.31\pm 5.6\times 10^{-3}$	22	±0.2535	±1.9222

**Table 2 sensors-23-03114-t002:** Comparative results of performance parameters while merging maps of same and different grid resolutions using MSAC and RANSAC algorithms.

ma	mb	MSAC Algorithm				RANSAC Algorithm			
$r_{a}$ (Cells/m)	$r_{b}$ (Cells/m)	Acceptance Index (ω)	|*Inliers*|	Rotation (deg.)	Translation (m)	Acceptance Index (ω)	|*Inliers*|	Rotation (deg.)	Translation (m)
25	25	$0.98\pm 2.3\times 10^{-4}$	33	±0.0472	±0.9274	$0.89\pm 1.7\times 10^{-4}$	28	±0.0512	±0.9881
20	10	$0.94\pm 3.2\times 10^{-3}$	22	±0.2535	±1.9222	$0.91\pm 1.4\times 10^{-3}$	19	±0.3451	±2.315

## Data Availability

The data related to this paper are available on request from the corresponding author.

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
