# Peer review of "Feature-Based Occupancy Map-Merging for Collaborative SLAM"

_sensors, 2023, doi:10.3390/s23063114_

Round 1

Reviewer 1 Report

This paper presents a hierarchical method to merge maps obtained from different robots by identifying similar key points in them. The proposed cooperative map merging algorithm can construct the global map even in case of low overlap between local maps and differences between in LiDAR resolution among robots.

1- Numeric description of some figures (e.g. Fig 8) are not visible. Font sizes should be increased.

2- Since the ultimate goal of map generation is navigation task, the authors should test their algorithm for map construction to evaluate robot path planning when the global map is constructed with similar and different LiDARs.

3- Authors compared the location of geometrically consistent features in Fig 7. It would be also informative to compare keypoints detectability of these feature extraction methods.  

4- The map fusion process is not clear enough. I recommend adding a pseudocode for map merging similar to RANSAC approach presented in Algorithm 1.

Reviewer 3 Report

Dear Authors,

In this paper, the authors present a feature-based map fusion approach for collaborative mapping that includes processing the spatial occupancy probabilities and detecting features based on locally adaptive nonlinear diffusion filtering.

What they also claim is

·         A procedure to verify and accept correct transformation.

·         Global grid fusion strategy, based on Bayesian inference which is independent of order of merging.

·         Identifying geometrically consistent features for low overlapping and various grid resolutions

·         Experimental results based on hierarchical map fusion to merge six maps at once.

Observations:

Overall, the paper is written very well. Yes, that’s true, the quality of the features identified, the precision of the predicted transformations, and the capability of the Bayesian inference method to appropriately combine the data from various maps are some of the elements that will determine whether such a strategy is successful.

The paper can be further improved by considering the following concerns:

·         How the state-of-the-art (FLIRT, FALKO, BID) key points detectors for 2D LIDAR data could be ignored when mentioning feature detection method for 2D data?

·         How the quality of features is better when the method involves an already existing approach?

·         In line 72-74, “proposed a fast and robust method of feature correspondences across various map conditions”. Firstly, please explain what novelty authors proposed in this method? Or is it just an implementation of the existing methods. Secondly, how do Figure 8(b) (for time) and figure (11) support this comment?

·         Figure 10 and 11 need better resolutions, additionally, a part of figure may also be zoomed in to show the merging features of the two maps in figure 10.

·         Why fusion step needed MATLAB? Does this time is also included in the time calculation step?

Regards;

Round 2

Reviewer 3 Report

My concerns on the previous version of the manuscript have been addressed sufficiently. Now, the paper paper is acceptable for possible publication.